# Paracellular Filtration Secretion Driven by Mechanical Force Contributes to Small Intestinal Fluid Dynamics

**DOI:** 10.3390/medsci9010009

**Published:** 2021-02-09

**Authors:** Randal K. Buddington, Thomas Wong, Scott C. Howard

**Affiliations:** 1School of Health Studies, University of Memphis, Memphis, TN 38152, USA; thomaslwong@gmail.com; 2Babies Taking Flight, Memphis, TN 38117, USA; 3Department of Acute and Tertiary Care, University of Tennessee Health Sciences Center, Memphis, TN 38163, USA; scott.howard@resonancehealth.org

**Keywords:** diarrhea, filtration secretion, pressure, tight junction, intestinal motility

## Abstract

Studies of fluid secretion by the small intestine are dominated by the coupling with ATP-dependent generation of ion gradients, whereas the contribution of filtration secretion has been overlooked, possibly by the lack of a known mechanistic basis. We measured apical fluid flow and generation of hydrostatic pressure gradients by epithelia of cultured mouse enterocytes, Caco-2 and T-84 cells, and fibroblasts exposed to mechanical force provided by vigorous aeration and in response to ion gradients, inhibitors of ion channels and transporters and in vitro using intact mouse and rat small intestine. We describe herein a paracellular pathway for unidirectional filtration secretion that is driven by mechanical force, requires tight junctions, is independent of ionic and osmotic gradients, generates persistent hydrostatic pressure gradients, and would contribute to the fluid shifts that occur during digestion and diarrhea. Zinc inhibits the flow of fluid and the paracellular marker fluorescein isothyocyanate conjugated dextran (MW = 4 kD) across epithelia of cultured enterocytes (>95%; *p* < 0.001) and intact small intestine (>40%; *p* = 0.03). We propose that mechanical force drives fluid secretion through the tight junction complex via a “one-way check valve” that can be regulated. This pathway of filtration secretion complements chloride-coupled fluid secretion during high-volume fluid flow. The role of filtration secretion in the genesis of diarrhea in intact animals needs further study. Our findings may explain a potential linkage between intestinal motility and intestinal fluid dynamics.

## 1. Introduction

The current understanding of the bidirectional movement of fluid into and out of the small intestine involves ATP-dependent generation of ion and solute gradients by the epithelium that drive osmotic flow through paracellular and transcellular pathways. The active uptake of glucose and other solutes [1,2,3] creates an inwardly directed gradient for water absorption. Despite unequivocal evidence for chloride-coupled fluid secretion [4,5], it is not universally accepted as the sole mechanism for the large-volume fluid shifts that occur during digestion of a meal and particularly during diarrhea [6,7,8].

An alternative concept known as filtration secretion emerged from in vitro and in vivo studies conducted 40 to 50 years ago. Movement of fluid into the lumen was increased by imposing pressure on the serosal surface, [9] increasing mesenteric venous pressure [10] or creating negative intraluminal pressure [11]. Stimulating contraction of the jejunal musculature to increase tissue fluid pressure also enhanced fluid secretion and decreased fluid absorption [12,13]. During processing of a meal, small intestine motility creates pressure differentials in the mucosa [14] that contribute to fluid secretion in vivo [15]. The relationship among motility, pressure differentials, and fluid secretion explains why hypermotility coincides with diarrhea. However, unlike chloride-coupled fluid secretion, the underlying mechanisms and regulation of filtration secretion have not been understood and the concept and contribution to small intestinal fluid dynamics have been overlooked.

While characterizing rheogenic glucose transport by low-resistance (“leaky”) epithelial monolayers prepared from freshly harvested murine enterocytes cultured on inserts with 3 µm pores [16], we were intrigued by the serendipitous discovery that when vigorous aeration was accidentally applied, fluid moved from the basolateral to the apical chamber, but not in the other direction, and resulted in a persistent hydrostatic pressure gradient. This unexpected observation led us to performing a series of eight studies using epithelia of cultured cells and intact small intestine to understand this phenomenon. We report herein a mechanism of fluid secretion driven by mechanical force that is independent of ion secretion and can be targeted for strategies to treat diarrhea and constipation. This mechanism does not diminish the role of chloride-coupled fluid secretion. Instead, our findings provide insights into the mechanism of filtration secretion that are consistent with the several liters of fluid that the intestine secretes during digestion and the even larger volumes during diarrhea, when there is gut hypermotility.

## 2. Materials and Methods

All phases of the research that used animals adhered to the NIH Guide for the Care and Use of Laboratory Animals and were approved by the University of Memphis Institutional Animal Care and Use Committee. C57Bl/6 mice, 8–12 weeks of age, were obtained from a breeding colony maintained at the University of Memphis. Sprague–Dawley rats were purchased from a commercial supplier (Envigo, Indianapolis, IN, USA). The mice and rats were maintained under standard environmental conditions and fed a standard laboratory rodent chow.

### 2.1. Preparation of Epithelial Monolayers Using Freshly Harvested Mouse Enterocytes

After a mouse was euthanized by CO_2_ asphyxiation, the entire small intestine from the pyloric sphincter to the ileocolonic junction was removed, flushed with cold mammalian Ringer’s solution, and everted. The enterocytes were harvested following the method of Kimura et al. [16]. Briefly, the everted intestines were incubated at room temperature for 10 min in Dulbecco’s Phosphate-Buffered Saline (DPBS) with 0.05% dithiothreitol and gentle agitation. After rinsing with room temperature DPBS, the intestine was incubated in a mixture of equal proportions of an enzyme-based dissociation/cell detachment solution (HyQTase^®^; HyClone; Logan, UT, USA) and an enzyme-free solution (Enzyme-free cell dissociation solution, PBS based; Millipore; Billerica, MA, USA) for 15–30 min at 37 °C with gentle agitation to release enterocytes from the villus epithelium. The suspension was filtered through cheesecloth, and the cells were sedimented, and rinsed three times with cold DPBS. This approach typically yields 15–30 million enterocytes per small intestine that were suspended at a density of 10 million enterocytes per ml in cold low-glucose Dulbecco’s modified Eagle’s medium (DMEM; Gibco, Life Technologies, Grand Island, NY, USA; Hyclone, Logan, UT, USA; Lonza, Allendale, NJ, USA) with 10% sterile filtered fetal bovine serum (FBS), 1 mL of a combination of penicillin (100 U/mL) and streptomycin (100 μg/mL), 5 mL of 4-(2-hydroxyethyl)-1-piperazineethanesulfonic acid (HEPES; 10 mM; pH = 7.3), and a commercial combination of insulin, transferrin, and selenium (ITS, added per supplier instructions; HyClone; Logan, UT or Lonza; Allendale, NJ, USA). The enterocytes were seeded onto uncoated 0.6 cm^2^ polycarbonate membrane inserts (Millipore, Billerica, MA) at a density of 1.5 million cells per insert and incubated for 7–9 h at 37 °C in 95% room air with 5% CO_2_.

The enterocytes harvested from each mouse intestine were used to prepare 6 or more inserts. The plated cells rapidly settled and within 6 h were reorganized into a polarized epithelium with distinct tight junctions, and with microvilli restricted to the outer surface. All of the studies were performed after 7 to 9 h of incubation while the cells remain viable and carrier-mediated glucose transport is maximal [16]. Like the intact small intestinal mucosa, the epithelia prepared from harvested enterocytes have low electrical resistance (120–130 uOhms/cm^2^) and are capable of rheogenic glucose uptake.

### 2.2. Culture of Caco-2 and T84 Cells and Fibroblasts

The majority of cell-based studies of intestinal chloride secretion use cell lines derived from the colon that form more extensive tight junctions, resulting in epithelia that have higher electrical resistance than the epithelia prepared from harvested enterocytes. We used Caco-2 (ATCC^®^ HTB-37™ Manassas, VA, USA) and T84 (ATCC^®^ CCL-248™) as representative colonic cell lines. We also cultured fibroblasts (Primary Dermal Fibroblast Normal; Human, Neonatal (ATCC^®^ PCS-201-010™ Manassas, VA, USA) because they do not form tight junctions. The Caco-2 and T84 cells and fibroblasts were seeded on uncoated 0.6 cm^2^ polycarbonate membrane inserts with 12 μm pores and cultured using the same media and conditions. Confluence was confirmed visually and the formation of the tight junction complexes by the Caco-2 and T84 cell was verified by increased resistance (300–500 Ohms/cm^2^ for Caco-2 and 1000–3000 Ohms/cm^2^ for T84 cells).

### 2.3. Measurement of Fluid Flow across the Epithelia

All of the studies using epithelia from harvested enterocytes used at least two separate sets of inserts for each condition with each set prepared from individual mice. The studies with Caco-2 and T84 cells used at least 3 inserts per condition. The inserts were mounted in an 8 chamber Ussing system (Physiological Instruments, San Diego, CA, USA) with the epithelium, and hence the apical side, exposed to the right chamber and the underside of the insert, hence the basolateral side and membrane, exposed to the left chamber. Placing inserts in the reverse orientation simply reversed the direction of flow.

Clear plastic rulers were attached to both chambers to record the movement of fluid in 0.5 mm increments. Unless otherwise specified, the two chambers were filled to the same level with 37 °C mammalian Ringer’s solution with (in mmol) 128 NaCl, 4.7 KCl, 2.5 CaCl_2_ 1.2 KH_s_PO4, 1.2 MgSO_4_, and 20 NaHCO_3_; 290 mosm; pH = 7.3–7.4 when aerated with 95% O_2_ with 5% CO_2_. Fluid flow was measured at 5 min intervals as the increase in fluid level in the apical chamber during which the chambers were aerated with a gas mixture of 95% O_2_ and 5% CO_2_. After the gas was turned off, the change in fluid level in the apical chamber (mm) was recorded and converted to a volume (mL) using a constant that was determined by measuring the volume of fluid per mm change in the height of fluid in the chamber. The fluid levels in the two chambers were equilibrated before starting the next measurement by removing the excess that had accumulated in the apical chamber and adding fresh Ringer’s solution to the basolateral chamber.

Dextran with a molecular weight of 4000 that was conjugated with fluoroscein isothiocyate (4 kD FITC-dextran) was added to either the apical or the basolateral chamber in some studies to visually track the movement of fluid.

### 2.4. Preparation of Sacs from Intact Mouse Small Intestine for Measurement of Fluid Flow

We considered it essential to determine whether the fluid flow by the cultured epithelial cells could be replicated using intact tissues. Additional C57/Bl6 mice from the same colony were used to prepare sacs of 4 to 6 cm length from small intestine segments that were either everted with the mucosa exposed or left non-everted with the mucosa internalized. The distal end was closed with a ligature and the entire length of intestine was filled using Ringer’s solution with 2 mg/mL 4 kD FITC-dextran until it expanded. Another silk ligature was used to close the proximal end. The sacs were gently blotted to remove adherent fluid, weighed, and then secured by the ligatures to metal rods. Throughout the preparation phase and after the sacs were attached to the rods, they were kept in cold (4 °C) Ringer’s solution aerated with a gas mixture of 95% oxygen and 5% CO_2_.

The measurement of fluid flow and leakage of 4 kD FITC-dextran was started by positioning the rods in an incubation tube such that the sacs were completely immersed in 20 mL of Ringer’s solution (37 °C) that was aerated with the gas mixture using a hematocrit tube. Additional agitation was provided by a stir bar rotating at ~1200 rpm. Samples (200 µL) of the incubation solution were collected at the start and at 10, 20 and 30 min and loaded onto a microplate to measure 4 kD FITC-dextran by recording fluorescence. After 30 min, the sacs were removed from the rods, gently blotted, and the ending weights were recorded. The difference between beginning and ending weight was used to estimate fluid loss.

### 2.5. Rat Intestine Mounted in the Ussing Chamber System

Other investigators using tissues mounted in Ussing systems remarked in discussions that they too had observed basolateral-to-apical fluid flow when aeration was mistakenly high (Dr. Steven Thompson, Physiologic Instruments, Inc., unpublished observations). We verified this observation using intestinal tissue that was obtained from two rats that were used in a behavior experiment and were euthanized by CO_2_ asphyxiation followed by decapitation. The use of rats also addressed a need to verify whether the fluid flow was shared by other species and not unique to mice. After only the serosa was removed, segments of jejunum were mounted in sliders with a 12.7 mm diameter circular aperture, exposing 1.27 cm^2^ of mucosa, and placed in the Ussing system. Both chambers were filled to the same level with Ringer’s solution and vigorous aeration was applied to both chambers for 30 min. After the volume of Ringer’s solution moving from the basolateral to the apical chamber was measured, the chambers were rebalanced, zinc was added to both chambers (to a final concentration of 100 μM), and the flow of fluid was again measured for 30 min. Both chambers were then drained, refilled with fresh Ringer’s solution, sodium citrate was added to both chambers (to a final concentration of 50 μM), and the volume of fluid flow was again measured for 30 min.

### 2.6. Statistics

Direct comparisons of paired values were made using *t*-tests. A one-way ANOVA was used for studies involving multiple comparisons. When a significant treatment effect was detected, specific differences among treatments were identified by Duncan’s multiple range test. For all tests, *p* < 0.05 was accepted as the critical level of significance. Values in figures are the means and standard errors.

## 3. Results

The following sections present the series of eight studies we performed to confirm the existence and understand this pathway of fluid flow driven by mechanical force. For each study, we describe the experimental approach and the associated findings.

### 3.1. Study 1. Epithelia of Mouse Enterocytes Establish Hydrostatic Pressure Gradients when Exposed to Vigorous Aeration

To confirm our initial observation of fluid flow, mouse enterocytes were cultured on 3 µm pore inserts and exposed to three different intensities of aeration; none, low (~5 mL/min typical of Ussing chamber studies), and vigorous (~500 mL/min); the same intensity of aeration was provided to both chambers. Ussing chambers are designed for the aeration to circulate the fluid, thereby exposing the epithelium to a current. Vigorous aeration increases the circulation of fluid, hence the current and the mechanical force imposed on the epithelium.

Two batches of enterocytes were mistakenly cultured on inserts with 0.4 µm pores, the size commonly used for culturing epithelia. Surprisingly, the resulting epithelia did not generate hydrostatic pressure gradients, even with 30 min of vigorous aeration. To determine whether there was an influence of insert pore sizes, we cultured mouse enterocytes on inserts with 0.4, 3, 8, and 12 µm pores and exposed them to the different intensities of aeration.

To understand whether the fluid movement was simply an artifact of insert pore size, inserts with different pores sizes, but without cells, were exposed to 30 min of vigorous aeration, with both chambers starting with the same level of fluid. We also explored the potential role of insert pore size as a barrier for passive fluid movement using cell free inserts. Both chambers were equilibrated with Ringer’s solution before adding an additional 0.5 mL to one chamber and recording the time for the levels in both chambers to re-equilibrate without aeration.

### 3.2. Findings

The rate of fluid movement through epithelia of cultured mouse enterocytes is directly related to the intensity of aeration and pore size (Figure 1). At low-intensity aeration, fluid flow was not detected when epithelia were cultured on inserts with 0.4 and 3 µm pores, whereas flow was evident, but low when epithelia were cultured on inserts with 8 and 2 µm pores. Without aeration, there was no fluid movement, regardless of pore size. There was no net fluid movement when vigorous aeration was applied for 30 min in a cell-free system, regardless of pore size and whether either or both chambers were aerated. These findings indicate that epithelia cultured on inserts with pores that are 3 µm and greater can establish hydrostatic pressure gradients when exposed to mechanical force caused by high-intensity aeration.

The flow of Ringer’s solution through inserts without cells was directly related to pore size (Table 1). This was explained by the Hagen–Poiseuille equation for fluid flow through a tube.
(1)$$\Phi \;=\;\frac{\pi R^{4}}{8\eta }\frac{\left|\Delta P\right|}{L}$$
where, in compatible units, Φ is the volumetric flow rate, *R* is the internal radius of the tube, which reflects pore size. Δ*P* is the pressure difference between the two ends, *η* is the dynamic fluid viscosity, and *L* is the length of the tube. The greater resistance of smaller tubes (pores) reduces flow. The hydrostatic pressure (Δ*P*) imposed by adding 0.5 mL to one chamber was equal for all inserts. *L* is represented by the thickness of the membrane and would be constant across the inserts, and viscosity (*ƞ*) was identical. If velocity of flow through the pores is assumed to be equal, then the volumetric flow rate (Φ) increases more than 6 orders of magnitude from 0.4 to 12 µm pores. This explains why the time required for the two chambers to re-equilibrate after the addition of 0.5 mL to one chamber decreased as pore size increased.

Further studies of fluid flow by epithelia prepared from enterocytes, Caco-2 and T84 cells, and fibroblasts used inserts with 12 µm pores and both chambers were aerated vigorously.

### 3.3. Study 2. The Movement of Fluid through Epithelia Is Unidirectional

The directionality of flow was ascertained before and after reversing the orientation of inserts with cultured mouse enterocytes and by aerating just the apical or just the basolateral chamber or both chambers. The fluid flow was visualized by adding 100 µL of a stock solution of 4 kD FITC-dextran (5 mg/mL), which is a marker of paracellular movement [17] to either the apical or the basolateral chamber before starting the vigorous aeration. After 5 min, a sample (200 µL) was taken from the opposite chamber to measure fluorescence along with the volume change.

We observed that the hydrostatic pressure gradient caused by the movement of fluid into the apical chamber did not diminish during the 2–3 min the aeration was turned off while we recorded the levels of fluid in the two chambers. The persistence of the hydrostatic pressure gradient was verified by exposing epithelia of mouse enterocytes to 5 min of vigorous aeration to create a hydrostatic pressure gradient then turning off the aeration and recording changes in volume after 15 min. As another indicator of the unidirectional flow, Ringer’s solution was added to the basolateral chamber to create a 10 mm basolateral-to-apical hydrostatic pressure gradient. Volume changes in both chambers in the absence of aeration were recorded for the next for 15 min.

### 3.4. Findings

Changing the orientation of the inserts did not change the basolateral-to-apical direction of flow. Vigorous aeration of only the apical compartment produced volume shifts similar to when both chambers were aerated, while aeration of only the basolateral chamber reduced the fluid shift by 50–60% (Figure 2). This is attributed to the insert membrane acting as a physical barrier that attenuates the mechanical force imposed on the epithelium caused by aeration of the basolateral chamber.

After adding the 4 kD FITC-dextran to the basolateral chamber, vigorous aeration caused rapid movement of both fluid and the FITC-dextran to the apical chamber (Figure 3A,B). Without aeration, the movement of 4 kD FITC-dextran from the basolateral to the apical chamber was reduced by >95%. When FITC-dextran was added to the apical chamber, there was minimal movement to the basolateral chamber, even with vigorous aeration (Figure 4). The unidirectional movement of fluid and FITC-dextran can be seen in the Appendix A.

The hydrostatic pressure gradients that were established by exposing epithelia of cultured enterocytes to vigorous aeration for 5 min (average of 9.5 mm rise in the apical chamber) did not decline during the subsequent 15 min period with no aeration (Figure 5). The imposed 10 mm basolateral-to-apical hydrostatic pressure gradient gradually diminished over 15 min without aeration. However, the volume of fluid passively moving into the apical chamber was <70% (*p* < 0.01) of the volume that moved during the 5 min of vigorous aeration. These findings provide further evidence the apical flow of fluid is unidirectional and is enhanced by aeration.

### 3.5. Study 3. Tight Junctions Are Required for the Paracellular Movement of Fluid

To ascertain the role of the tight junction, fluid flow was measured by Caco-2 and T84 cells that develop extensive tight junctions and are commonly used as models of intestinal epithelia and fibroblasts that form monolayers while lacking many tight junction proteins.

### 3.6. Findings

Exposure of epithelia of Caco-2 and T84 cells to vigorous aeration confirmed the movement of fluid into the apical chamber is unidirectional, but the volume was <20% of what was measured using epithelia of harvested enterocytes (Figure 6) corresponding with higher epithelial resistance. Net fluid movement was not detected in either direction when inserts with fibroblasts were exposed to vigorous aeration. Hence, tight junctions appear to be necessary for the paracellular fluid movement and the magnitude of flow may be related to the extent of tight junction development.

### 3.7. Study 4. Aeration Drives Fluid Flow Independent of Ion Movements or Gradients

The accepted coupling of chloride and fluid secretion led us to use multiple approaches to determine whether the flow of fluid driven by aeration was coupled with ion secretion. We prepared chloride-free Ringer’s solution by substituting sodium gluconate and potassium gluconate for NaCl and KCl. The chloride-free Ringer’s solution was added to the basolateral chamber with the expectation a basolateral-directed chloride gradient would at least slow and perhaps even reverse the flow of fluid by epithelia of cultured mouse enterocytes and Caco-2 cells. A second prediction was that an apical-directed chloride gradient by adding the chloride-free Ringer’s solution to the apical chamber would increase the flow of fluid. We also exposed epithelia of mouse enterocytes to gradients for sodium and potassium that were established using choline chloride for NaCl and KCl and choline bicarbonate for NaHCO_3_. Another study assessed the possible role of bicarbonate secretion by measuring fluid flow while aerating with 100% nitrogen.

Because cell viability and epithelial integrity might be affected by prolonged exposure to Ringer’s solution devoid of specific ions, fluid flow by cultured enterocytes was measured in the presence of various compounds known to alter membrane fluxes of ions (Table 2). These were added to normal Ringer’s solution at concentrations known to inhibit ion fluxes first to each chamber separately and then to both chambers.

As a definitive experiment, we created an osmotic gradient by filling the apical chamber with Ringer’s solution that had been diluted with water to 200 mOsm (69% of normal). The expectation was that this osmotic gradient would slow the flow of fluid into the apical chamber and possibly reverse the flow into the basolateral chamber.

### 3.8. Findings

Even though the enterocytes used to prepare the epithelia are from the upper villus [16] and express CFTR [18], creating a basolateral-directed chloride gradient did not reduce the unidirectional movement of fluid into the apical chamber (Figure 7A). Conversely, adding chloride-free Ringer’s solution to the apical chamber to create an apical-directed chloride gradient did not increase the rate of fluid flow. Addition of inhibitors of chloride channels and transporters did not diminish the fluid flow (Table 2). Therefore, the flow of fluid associated with vigorous aeration was independent of chloride.

Interestingly, when Caco-2 cells that are able to secrete chloride [19] were exposed to an apical-to-basolateral chloride gradient, aeration reversed the direction of fluid movement (Figure 7B). These findings suggest that epithelia derived from colonocytes respond differently to the combination of vigorous aeration and chloride gradients than epithelia prepared using enterocytes.

The slight reversal of fluid flow when epithelia prepared from Caco-2 cells were exposed to chloride-free Ringer’s solution in both chambers led us to consider whether other ions may be involved. Establishing sodium or potassium gradients did not alter the flow of fluid by epithelia prepared from enterocytes, irrespective of whether the choline substituted solutions were added to the apical, basolateral, or both chambers (Figure 8A), nor did the addition of ouabain to inhibit Na-K-ATPase alter fluid flow (Table 2). Moreover, the fluid flow was not lower with vigorous aeration using 100% nitrogen to reduce bicarbonate production compared with the mixture of oxygen and carbon dioxide (Figure 8B). The basolateral-to-apical fluid movement was not affected by substituting cesium for the calcium and magnesium or by adding the various inhibitors of CFTR and other chemicals known to inhibit transcellular and transmembrane ion movements, when added alone and in combinations to both chambers. 

An even more striking indicator of the ion and osmotic independence of filtration secretion is how flow from the basolateral to the apical chamber was not hindered by filling the apical chamber with diluted Ringer’s solution to impose a counter osmotic gradient (Figure 9).

### 3.9. Study 5. Fluid Flow Driven by Mechanical Force Is Not Directly Dependent on Cellular Metabolism

Active ion secretion requires energy and is temperature sensitive [20]. This led us to predict that fluid flow by epithelia would be temperature sensitive. This was tested by measuring fluid flow first at 10 °C, then at 20 °C, and then at 37 °C. To ascertain the role of ATP, we compared rates of fluid flow at 5 min intervals for 30 min with and without the presence of the ATP diphosphohydrolase, apyrase (10 U/mL) to deplete extracellular ATP. After learning there was a gradual, but significant decline in flow with multiple measurements by the third recording (see below), we determined whether the significant decline in flow during the first 15 min would be reduced by adding ATP, AMP, or adenosine to both chambers (at 0.5 mM).

### 3.10. Findings

Fluid movement at 10 °C (0.58 ± 0.004 mL/min·cm^2^) was surprisingly higher (*p* < 0.05) than subsequent measurements made with the same inserts at 20 °C (0.48 ± 0.005 mL/ min·cm^2^) and 37 °C (0.41 ± 0.004 mL/min·cm^2^).

The rate of fluid movement at 37 °C gradually declined with repeated measurements, suggesting the decline with increasing temperature may have been partly caused by the repeated measurements. The decline was more rapid when extracellular ATP was depleted using apyrase (Figure 10A). Because of these findings all future studies of fluid flow using the same inserts were restricted to 3 or fewer consecutive recordings. Adding ATP, AMP, or adenosine to both chambers reduced the rate of decline at 15 min (Figure 10B), perhaps by activation of P2Y purinergic receptors and altering tight junction characteristics [21].

### 3.11. Study 6. Zinc Inhibits Fluid Flow by Epithelia

Regulation of fluid movement through the small intestine epithelium is of physiological interest and clinical relevance. We selected zinc as a candidate inhibitor of fluid flow because of its purported antidiarrheal properties [22,23,24,25,26] and influence on epithelial tight junction integrity and ion secretion [27,28,29,30,31]. Fluid flow by cultured enterocytes exposed to vigorous aeration was first measured for 5 min without zinc to obtain a baseline rate of flow. After fluid levels in both chambers were re-balanced, fluid flow was again measured after adding ZnSO_4_ to either the apical or basolateral chamber or to both chambers to a final concentration of 100 µM, a concentration sufficient to alter epithelial characteristics in various experimental models [27,31,32,33]. We next defined a dose–response relationship for the extent of inhibition using increasing concentrations of ZnSO_4_; both chambers received the same concentration. To allay potential concern that the inhibition was caused by sulfate, we exposed epithelia to 100 µM concentrations of zinc chloride and zinc nitrate. We also ascertained whether the inhibition is unique to zinc or is shared with other transition metals by measuring the response of fluid flow to the addition of 100 µM copper sulfate, cesium sulfate, iron sulfate, and calcium sulfate.

To determine whether zinc damages the tight junctions and the epithelium, which would prevent a hydrostatic pressure gradient, zinc was added to both chambers and additional Ringer’s solution was added to the apical chamber to create a 0.5 cm hydrostatic pressure gradient and the change in volume was observed for 5 min. The prediction was the imposed hydrostatic pressure gradient would dissipate if zinc damaged the tight junctions or epithelium.

After inhibiting fluid flow by adding zinc sulfate or copper sulfate, it was important to determine whether the flow could be restored. Citrate was selected for evaluation because it chelates zinc and copper and is included in many bowel preparation solutions. The hypothesis was chelation of the zinc (and copper) by citrate (50 μM) would restore flow. However, citrate also chelates calcium, though not as avidly as EDTA and this could impact the integrity of tight junctions. Therefore, we also determined whether citrate alone would affect fluid flow.

### 3.12. Findings

Zinc sulfate (ZnSO_4_) caused an immediate inhibition of fluid movement that was dose dependent (Figure 11A) and the inhibition was more pronounced when ZnSO_4_ was added to the basolateral chamber (Figure 11B), whereas addition to the apical chamber was less effective; subsequent addition of ZnSO_4_ to the basolateral chamber caused full inhibition. The ability of the epithelium to prevent backflow was not compromised by zinc (Figure 11C), indicating the cessation of net flow by zinc was not caused by damage to epithelial integrity.

The magnitude of inhibition caused by ZnSO_4_ (>95% of initial fluid flow when all of the studies were pooled) was matched by adding zinc chloride and zinc nitrate, indicating the inhibition was by zinc, not sulfate. Of the other metal ions evaluated, only copper sulfate inhibited fluid movement, as previously reported [25], and the 89% inhibition was comparable to the various forms of zinc.

The inhibition of fluid movement by ZnSO_4_ and CuSO_4_ was restored by adding citrate (Figure 11D). This indicates that neither zinc nor copper irreversibly disrupts the tight junction or damages the epithelium. Adding citrate without any prior inhibition caused an insignificant increase in fluid flow compared to the baseline (110% ± 5% of fluid flow before addition of citrate). If citrate damages the tight junctions or the epithelium, this will instead cause a decrease in fluid flow.

### 3.13. Study 7. Agitating Sacs of Mouse Small Intestine Causes Fluid Flow

Because proximal-to-distal gradients are documented for many small intestinal functions, we first compared fluid loss based on weight changes and FITC-dextran leakage by sacs prepared from the proximal 50% and the distal 50% of the small intestine and suspended in test tubes with Ringer’s solution that was intensively aeration and stirred using a stir bar rotating at 1200 rpm. We then examined the role of mechanical force by comparing fluid loss and 4 kD FITC-dextran leakage by adjacent sacs of everted intestine exposed to vigorous aeration and stirring or very gentle aeration and no stirring. We also compared the loss of fluid and 4 kD FITC-dextran by sacs prepared from adjacent segments that were either everted to expose the mucosa to agitation by the aeration and stirring or not everted so the mucosa was internal and not directly exposed to agitation.

We next compared weight loss and 4 kD FITC-dextran leakage by sacs of everted mouse intestine that were incubated in vigorously aerated and stirred Ringer’s solution with and without ZnSO_4_ (100 µM).

### 3.14. Findings

Everted sacs prepared from the proximal small intestine exposed to intense aeration and stirring lost 33.9 ± 5.5% of the original weight. This was not significantly different (*p* = 0.11) than the decrease in weight by sacs prepared from the distal small intestine (20.7 ± 5.4%). The respective losses in weight coincided with an insignificant difference in the leakage of 4 kD FITC-dextran based on leakage into the bath solution (353 ± 62 fluorescent units/min versus 252 ± 51 for proximal and distal intestine, respectively; *p* = 0.23). To avoid any potential regional influences all subsequent comparisons were made using sacs prepared from adjacent segments of small intestine.

Sacs from small intestine segments that were everted with the mucosa exposed lost more weight and leaked more 4 kD FITC-dextran than sacs that had not been everted with the mucosa internal and not directly exposed to the agitation caused by the aeration and stirring (Figure 12A). Similarly, everted sacs incubated in static conditions (no stirring and gentle aeration) lost less weight and 4 kD FITC-dextran than everted sacs exposed to vigorous agitation. If the loss of fluid was driven by a hydrostatic pressure gradient caused by filling the sacs with Ringer’s solution, then agitated and non-agitated sacs would have lost similar volumes of fluid and 4 kD FITC-dextran. Hence, exposing the mucosa to agitation (mechanical force) increases the trans-tissue movement of fluid and 4 kD FITC-dextran toward the luminal face, corroborating our findings for epithelia prepared from mouse enterocytes.

Adding ZnSO_4_ to the incubation solution decreased the loss of weight and leakage of 4 kD FITC-dextran by everted sacs exposed to vigorous aeration and stirring (Figure 12A). This differs from the complete inhibition of fluid flow when epithelia are exposed to zinc. The modest loss of weight and leak of 4 kD FITC-dextran by non-everted sacs with the mucosa internal and not exposed to agitation also differs from the lack of fluid flow and 4 kD FITC-dextran loss by epithelia of mouse enterocytes exposed to stagnant conditions. This may be caused by mucosal absorption counteracting any fluid flow into the lumen.

### 3.15. Study 8. Fluid Flow by Intact Rat Small Intestine Mounted in Ussing Chambers

The full thickness sections of rat intestine mounted in sliders and placed in the Ussing chambers were used as a second in vitro assessment of fluid flow and the response to zinc. The movement of fluid in response to intense aeration was measured before and after adding ZnSO_4_ and again after adding citrate.

### 3.16. Findings

Exposure of intact rat intestine to vigorous aeration resulted in movement of fluid from the basolateral to the apical chamber (Figure 12B). The movement of fluid was inhibited by zinc sulfate and was largely restored by citrate.

## 4. Discussion

Diarrheal diseases are a leading cause of childhood malnutrition and each year, the 1.7 billion cases of childhood diarrheal disease worldwide kill more than 525,000 children younger than 5 years old [26]. Inhibiting the excessive secretion of fluid is essential. Our results provide novel insights into small intestinal fluid dynamics and importantly how gut hypermotility during diarrhea can contribute to the rapid secretion of high volumes of fluid and why the inclusion of zinc in oral rehydration solutions is efficacious [26].

Our initial observation that cultured enterocytes exposed to vigorous aeration establish hydrostatic pressure gradients led us to perform the series of eight studies that demonstrated the fluid movement was not an artifact of the experimental paradigm (Table 3). Instead, mechanical force, provided by agitation of the bath solution in this paradigm, drives unidirectional movement of fluid through intestinal epithelia that is lumenally directed, paracellular, independent of ion secretion, capable of generating a hydrostatic pressure gradient, and can be reversibly regulated. We did not detect fluid flow when epithelia were cultured on inserts with 0.4 µm pores that have low fluid permeability. Since most Ussing chamber studies use inserts with this pore size and apply gentle aeration, it is not surprising that the ability of intestinal epithelia to establish and maintain hydrostatic pressure gradients has not been previously reported.

Our in vitro demonstration of fluid flow using different preparations of intact mouse and rat small intestine is consistent with the in vivo 30-fold higher serosal to mucosa movement of high molecular weight markers compared with low mucosal to serosal movement [34]. Our findings implicating mechanical force in driving fluid movement correspond with early descriptions of increased filtration secretion by intact small intestine in response to imposed pressure on the serosal surface [9], increased venous pressure [10], negative intralumenal pressure [11], and importantly by contraction of the smooth muscle [12,13]. This pathway of fluid secretion driven by mechanical force complements chloride-coupled fluid secretion and contributes to a better understanding of small intestinal fluid dynamics, and particularly the increased secretion of fluid during digestion of meals when gut motility is stimulated [14] and the massive volumes secreted during diarrhea.

### 4.1. The Evidence for Ion-Coupled Fluid Secretion

In vivo studies using cholera toxin to activate the cystic fibrosis transmembrane regulator (CFTR) linked the accumulation of fluid in isolated segments of small intestine with in vitro Ussing chamber measurements of increased flux of chloride into the apical (luminal) compartment [35]. The coupling of chloride and fluid secretion was further supported by a mouse model of cystic fibrosis (CF) that is resistant to secretory diarrhea, and cholera toxin causes a reduced short circuit current and less fluid accumulation in isolated segments [36]. Similarly, inhibitors of CFTR and the signaling pathways that activate CFTR reduce fluid accumulation by intact segments of intestine from normal mice [4]. The gradual swelling of organoids composed of differentiated intestinal cells in response to CFTR activation [37] provided additional evidence for chloride-mediated fluid secretion. Collectively, these findings have focused the development of anti-diarrheal drugs on inhibiting CFTR-associated chloride fluxes [38,39]. Unfortunately, despite decades of clinical development effort, inhibitors of CFTR have not proven successful at treating diarrhea [6].

### 4.2. Fluid Secretion Can Be Independent of Ion Secretion

With research focused on chloride-coupled fluid secretion, the contribution of filtration secretion has been overlooked. The ability of cultured enterocytes to generate hydrostatic pressure gradients when exposed to vigorous aeration was not prevented by inhibitors of ion channels and transporters nor was it diminished by imposing ion and osmotic gradients that should have reversed the flow if solely reliant on ion movements. Even low temperature did not hinder fluid flow, despite the temperature sensitivity of ion channels and transporters [20]. Our findings demonstrate that fluid movement can occur independent of ion secretion. Clinical examples of ion-independent fluid secretion are the diarrheas caused by Shigella toxin [40,41] and Giardia [42]. Our findings indicate that gut hypermotility caused by Shigella toxin and Giardia drives fluid secretion into the gut and along with reduced electrolyte and fluid absorption plays a central role in causing the diarrhea.

The reversed apical-to-basolateral flow when epithelia of Caco-2 cells were exposed to a basolateral-directed chloride gradient and vigorous aeration reinforces that the phenomenon is not an artifact. If the unidirectional basolateral-to-apical flow was caused by the epithelium covering the pores and preventing backflow, the reversal of flow would not occur. The contrasting responses to chloride gradients by cultured enterocytes and Caco-2 cells derived from the colon may reflect the different origins and associated characteristics.

### 4.3. Mechanical Force Contributes to Small Intestinal Fluid Secretion

There are several sources of anecdotal evidence supporting a role of mechanical force in causing fluid secretion by the intestine. While investigating the influence of mechanical vibrations on organisms and possible applications for medical therapies, Nicolas Tesla made the following observation.

“…some of us, who had stayed longer on the (vibrating) platform, felt an unspeakable and pressing necessity which had to be promptly satisfied.”(Nicolas Tesla and Mechanical Therapy, 1896).

The “necessity” referred to by Tesla was diarrhea. What he did not understand is that the vibrating platform caused diarrhea by disturbing the carefully regulated dynamic balance between the counter processes of fluid secretion and absorption that determines the fluid volume in the gut lumen. From the present results, we infer the vibrations imposed an external mechanical force on the intestine that caused filtration secretion and a large volume of fluid to rapidly flow into the small intestine. The link between mechanical force and fluid secretion explains why whole-body vibration [43] and vibrating capsules [44,45] improve stool consistency and frequency among patients with functional constipation. Additional evidence linking movement of the gut with fluid secretion is exercise-induced diarrhea, which is a common malady among endurance runners [46], but is rare among endurance cyclists and swimmers, whose events impose much less jostling of the gut [47].

Contractions of the smooth muscle during the processing of a meal impose a physiological mechanical force on the intestinal mucosa and epithelium [14]. The present findings suggest that this force contributes to fluid secretion [15]. Increased perfusion of the intestinal vasculature is another driving force for filtration secretion of fluid into the intestine [10,48] and would be concomitant with increased motility. Although present in vivo, it was absent in the cell and intact intestinal tissue preparations we used. Our findings imply that gut hypermotility associated with diarrhea may contribute to the secretion of large volumes of fluid and explains why anti-diarrheal agents that inhibit smooth muscle contraction remain the most effective [49].

Another consideration are the energetic costs and oxygen demands for basal chloride secretion are significant [50], likely excessive for the ~3 L of fluid normally secreted each day by the human intestine during digestion, and even more during diarrhea [51]. From an evolutionary perspective, coupling increased intestinal motility during the processing of meals with the secretion of higher volumes of fluid would be energetically favored. The small intestine hypermotility during severe diarrhea is a more likely contributor to for the 20 L or more of fluid lost per day than chloride secretion alone. This is consistent with inhibitors of chloride secretion for treatment of diarrhea have not proven effective clinically in contrast to anti-diarrheal agents that inhibit motility. This explains why CFTR-focused therapies that decrease smooth muscle contraction have shown promise as anti-diarrheal agents [52].

### 4.4. A Combination of Ion Secretion and Mechanical Force Contribute to Small Intestinal Fluid Secretion

The relative contributions of ion secretion and motility for small intestine fluid secretion need to be established for normal and disease states. Our findings suggest that during periods of high-volume fluid secretion, such as during digestion and diarrhea, filtration secretion driven by the increased motility plays a major role, whereas CFTR and chloride-coupled fluid secretion will be important between meals when the small intestine is mostly quiescent. An important consideration is that smooth muscle cells express CFTR and the state of contraction is responsive to activation and inhibition of CFTR [53]. CF patients and a mouse model of CF have reduced intestinal motility [54,55] that contributes to the intestinal dysfunctions of ileus and constipation that are associated with CF in human subjects [56] and animal models of CF [57]. The various functional GI disorders associated with channel defects, as exemplified by cystic fibrosis [58] provide evidence of how a combination of ion secretion and motility contribute to the secretory component of intestinal fluid balance.

Both chloride secretion and smooth muscle contraction are stimulated by cholera toxin [59], by 5-hydroxytryptamine released by rotavirus infection [60] and by gentle stroking of the mucosa [61,62], another form of mechanical force. However, intestinal mucosa mounted in Ussing chambers is usually stripped of the muscle layers and exposed to gentle aeration. Despite increased chloride secretion in response to secretogogues, these preparations do not develop hydrostatic pressure gradients, corresponding with the lack of an imposed mechanical force needed for sufficient filtration secretion. Our exposure of unstripped intact small intestine mucosa to agitation by aeration and stirring caused measurable fluid movement and established hydrostatic pressure gradients. Because the accumulation of fluid by isolated sacs from normal animals exposed to cholera toxin has not been examined with inhibition of smooth muscle contraction [63], the contribution of filtration secretion has been overlooked.

Our findings provide an explanation for the fact that the most effective therapeutic agents for diarrhea and constipation are known to alter both chloride secretion and motility. These includes 5-hydroxytryptamine receptor 4 agonists and other prokinetic agents [64,65], prosecretory drugs, such as lubiprostone [66,67,68,69], and high concentrations of bile acids, such as deoxycholic acid [70]. The relationship between gut motility and fluid secretion provides insights into how prokinetic drugs can contribute to increasing stool moisture content and volume, address the constipation caused by CF, and have the potential to cause diarrhea and fluid loss. Conversely, therapeutic agents for diarrhea that inhibit motility can also reduce chloride secretion. Coinciding with this, there is interest in identifying and targeting intestinal receptors that regulate secretion and motility [71,72,73]. Currently underexplored is whether drugs targeting ion secretion and motility influence tight junction characteristics [74].

### 4.5. The Role of the Tight Junction in Unidirectional Fluid Flow

Although aquaporins are implicated in intestinal fluid dynamics [5], they are not considered the principal pathway of fluid flow into the lumen. The unidirectional movement of the paracellular marker 4 kD FITC-dextran [17] along with fluid indicates that filtration secretion involves a paracellular pathway but does not rule out that a portion of fluid movement is transcellular. The paracellular pathway is consistent with the tight junction complex including a regulated water channel [17] that we now propose allows for unidirectional fluid flow. The different rates of flow among epithelia prepared with enterocytes compared with Caco-2 and T-84 cells derived from the colon, and between the proximal and distal small intestine correspond with regional differences in the expression of claudins and other tight junction proteins [75,76], and perhaps the linkages with intracellular elements of the cytoskeleton [77] that are determinants of tight junction characteristics, epithelial permeability, and responses to different conditions and signals. It is unclear whether and how the unidirectional movement of fluid through the tight junction into the small intestine during the digestion of a meal would affect the paracellular absorption of nutrients.

### 4.6. Zinc, the Tight Junction, and Fluid Flow

Our discovery that filtration secretion can be reversibly regulated is a key finding and is relevant scientifically and important clinically. The inhibition of fluid and 4 kD FITC-dextran movement caused by zinc, even with vigorous aeration and agitation, is consistent with increased epithelial resistance and tight junction barrier functions [27,28,29,30,31,32]. The rapid decrease in flow may involve the zinc-sensing receptor, ZnR/GPR39 [78,79], and associated signaling pathways [80] triggering changes in tight junction characteristics, reducing electrogenic ion transport responses to secretogogues [27,29,31,81] and the magnitude of compromised epithelial barrier functions caused by *Salmonella* infection [82] and seminal fluid [83].

The proportionally greater basolateral inhibition compared with apical exposure is consistent with other studies using cultured cells and tissues [25,27,28,29,30,84]. However, the reduced fluid loss and 4 kD FITC-dextran leak when the mucosa of everted sacs of small intestine was exposed to zinc suggests that apical delivery in vivo may still be effective and explains the anti-diarrheal benefits of oral zinc supplements [85] and inclusion in oral rehydration solutions [26]. The greater magnitude and polarity of zinc inhibition of fluid flow by the epithelia are unexplained but provide a sensitive model to elucidate the mechanism of regulation.

The rapid restoration of fluid flow with citrate, though not immediate or complete, indicates that the metal ions did not disrupt the tight junctions. Specifically, if zinc and copper caused the loss of tight junction integrity, the epithelia would have been incapable of establishing and maintaining hydrostatic pressure gradients. Moreover, the rapid restoration of fluid flow by citrate suggests that this pathway of fluid flow into the intestine can be considered as a target for therapies to address diarrhea as well as constipation.

### 4.7. A New Model of High-Volume Intestinal Fluid Secretion Driven by Mechanical Force

We propose that filtration secretion and high-volume fluid flow into the lumen of the small intestine involve a “one-way check valve” as a component of the tight junction complex (Figure 13). Similar to a simple bellows, compression of the intercellular space by contraction of smooth muscle in the villus and muscle layers forces fluid through the tight junction into the lumen. The proposed one-way check valve likely exists in an open state and two closed states, as proposed for other paracellular channels [86]. In the open state (Figure 13A), fluid in the intercellular space is forced through the valve into the lumen. In the transient closed state (Figure 13B), the valve is closed because of the lack of a driving force and in this configuration prevents back flow from the lumen. This allows fluid to accumulate in the lumen, as seen with intestinal sacs [35,36], and create a persistent hydrostatic pressure gradient (present study). In the presence of a critical, yet unknown in vivo concentration of zinc and perhaps other ions and molecules, the valve adopts a stable closed (inactive) state, and fluid is prevented from flowing in either direction (Figure 13C). This hypothetical model is consistent with the increased barrier functions of the tight junction in response to zinc and other molecules [32]. Further studies are needed to define the structural, functional, and regulatory characteristics of this heretofore unknown unidirectional paracellular fluid pathway. Importantly from a global health perspective, the reversible inhibition by zinc and perhaps other ions or molecules provides a novel and promising target for therapeutics to treat diarrhea and constipation.

## 5. Conclusions

Although our findings contribute to a better understanding of the linkage between intestinal motility and filtration secretion, additional research is needed to address unanswered questions. Of particular importance is understanding how in vitro fluid secretion pertains to intestinal fluid dynamics in normal and disease states. There is uncertainty about how the experimental paradigm of agitating the bath solution corresponds with the mechanical force imposed on the epithelium in vivo by intestinal motility. There is a need to ascertain the role of the tight junction in filtration secretion, identify associated tight junction components and obtain morphological and molecular evidence for the proposed model, and elucidate the regulatory mechanisms underlying how zinc and other modulators of filtration secretion alter the rate of fluid movement. An important consideration for future studies is determining the relative contributions of filtration secretion and chloride-coupled secretion to intestinal fluid dynamics at different intensities of motility.

## Figures and Tables

**Figure 1 medsci-09-00009-f001:**
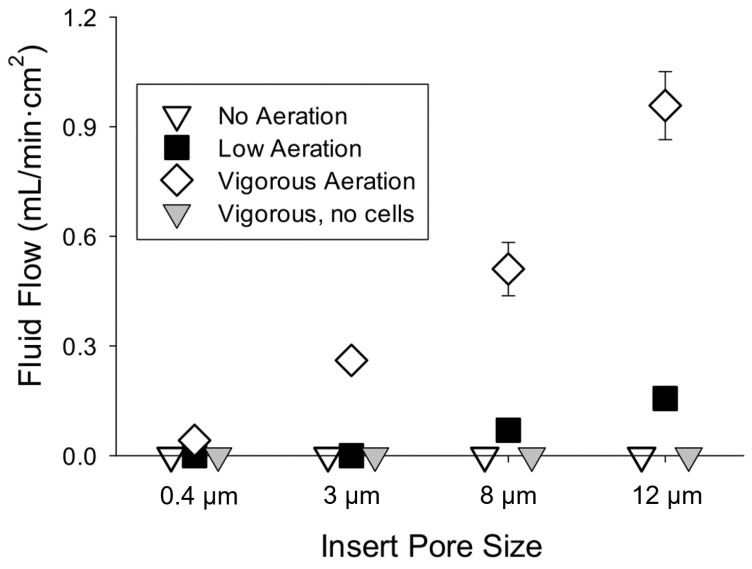
Fluid flow through epithelia prepared from mouse enterocytes and cell-free inserts and the relations with insert pore size and intensity of aeration. Enterocytes harvested from three mice were cultured on inserts with different pore sizes (N = 6 per size and aeration intensity). Fluid movement was measured as the volume increase in the apical chamber after exposure for 5 min to different intensities of aeration in both chambers.

**Figure 2 medsci-09-00009-f002:**
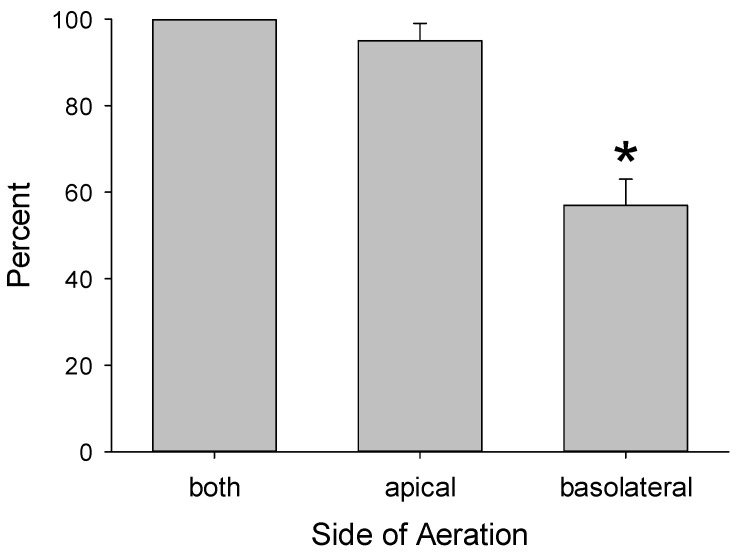
Fluid flow by mouse enterocytes cultured on inserts with 12 µm pores was measured when only the apical and then only the basolateral chamber was vigorously aerated. Values are expressed as the percentages of the fluid flow when both chambers were aerated (*n* = 6 inserts with three inserts from each of two preparations of mouse enterocytes). The asterisk indicates that aeration of only the basolateral chamber resulted in lower fluid flow compared with aeration of the apical or both chambers, which were equal.

**Figure 3 medsci-09-00009-f003:**
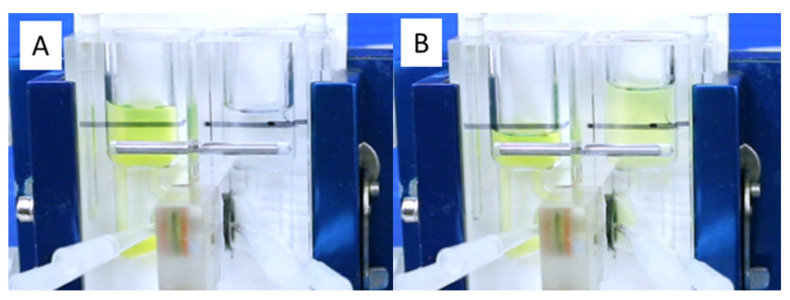
Photos immediately after 4 kD FITC-dextran was added to the basolateral chamber (**A**) and after 5 min of vigorous aeration (**B**) showing the movement of both fluid and FITC-dextran from the basolateral to the apical chamber and establishment of a hydrostatic pressure gradient.

**Figure 4 medsci-09-00009-f004:**
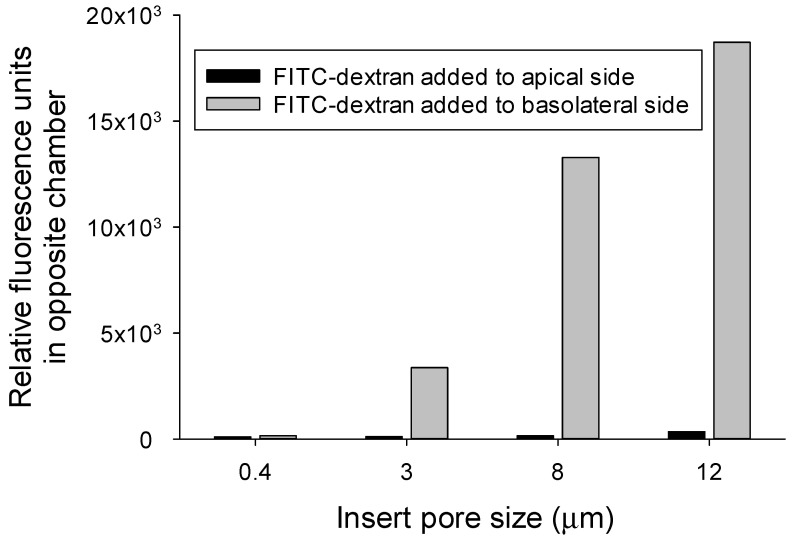
Mouse enterocytes were cultured on inserts with 0.4, 3, 8, and 12 µm pores. The directionality of FITC-dextran movement was evaluated by adding FITC-dextran to either the apical or to the basolateral chamber before starting 5 min of vigorous aeration and measuring fluorescence accumulation in the other chamber. (*n* = 6 inserts for each value).

**Figure 5 medsci-09-00009-f005:**
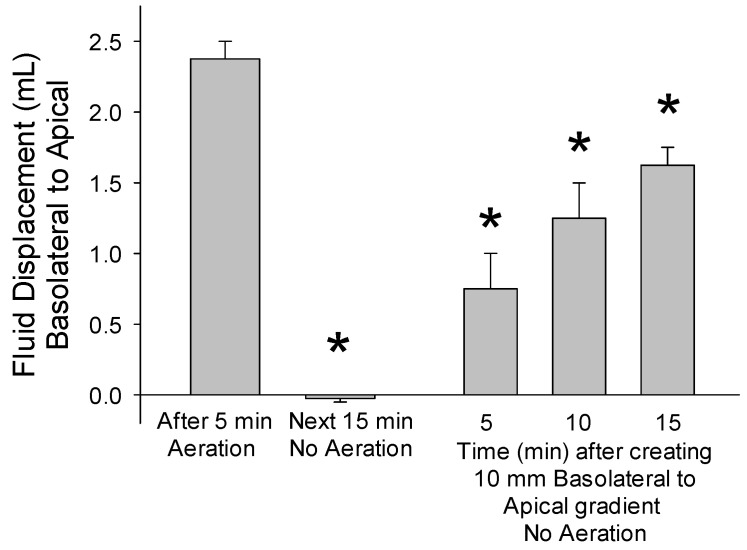
Mouse enterocytes were plated on inserts with 12 µm pores. After an apical-to-basolateral hydrostatic pressure gradient was established by 5 min of vigorous aeration (first bar), flow back to the basolateral chamber during the next 15 min, which would be negative, was minimal (second bar). The basolateral-to-apical gradient that was made by adding excess fluid to the basolateral chamber gradually diminished (bars 3–5). Asterisks indicate *p* < 0.01 for comparison with first bar for the volume of basolateral-to-apical fluid movement caused by 5 min of vigorous aeration (*n* = 6 inserts for each bar originating from multiple mice).

**Figure 6 medsci-09-00009-f006:**
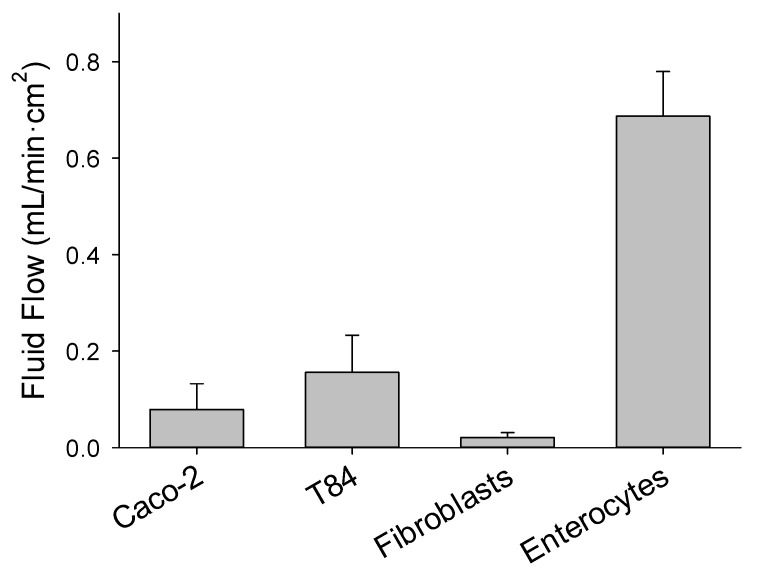
The rate of fluid flow was measured by epithelia prepared with Caco-2, T84 cells, fibroblasts, and harvested enterocytes that were cultured on 12 µm insert and exposed to vigorous aeration for 5 min.

**Figure 7 medsci-09-00009-f007:**
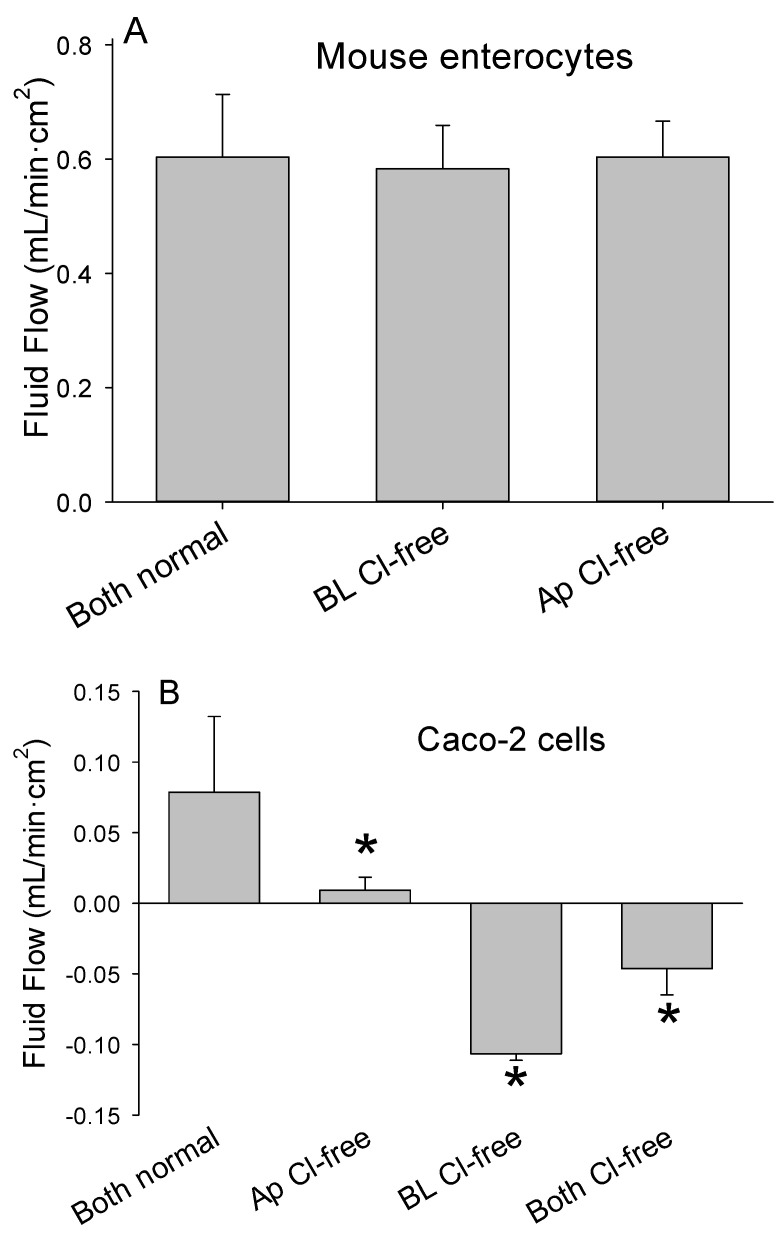
(**A**) Enterocytes harvested from two mice were cultured on inserts with 12 µm pores (*n* = 6 per mouse). Fluid flow to the apical chamber after 5 min of vigorous aeration did not differ when measured with normal Ringer’s solution in both chambers, or when chloride-free Ringer’s solution was added to the basolateral (BL Cl-free) or to the apical (Ap Cl-free) chamber. (**B**) Caco-2 cells were cultured on inserts with 12 µm pores (*n* = 8). Fluid moved from the basolateral to the apical chamber (positive value) in response to vigorous aeration with normal Ringer’s solution in both chambers (both normal), though significantly less than cultured enterocytes. Fluid movement to the apical chamber was reduced when Cl-free Ringer’s solution was added to the apical (Ap Cl-free) and actually reversed when added to the basolateral (BL Cl-free) or to both chambers (both Cl-free). * *p* < 0.05 for comparison with normal Ringer’s solution in both chambers.

**Figure 8 medsci-09-00009-f008:**
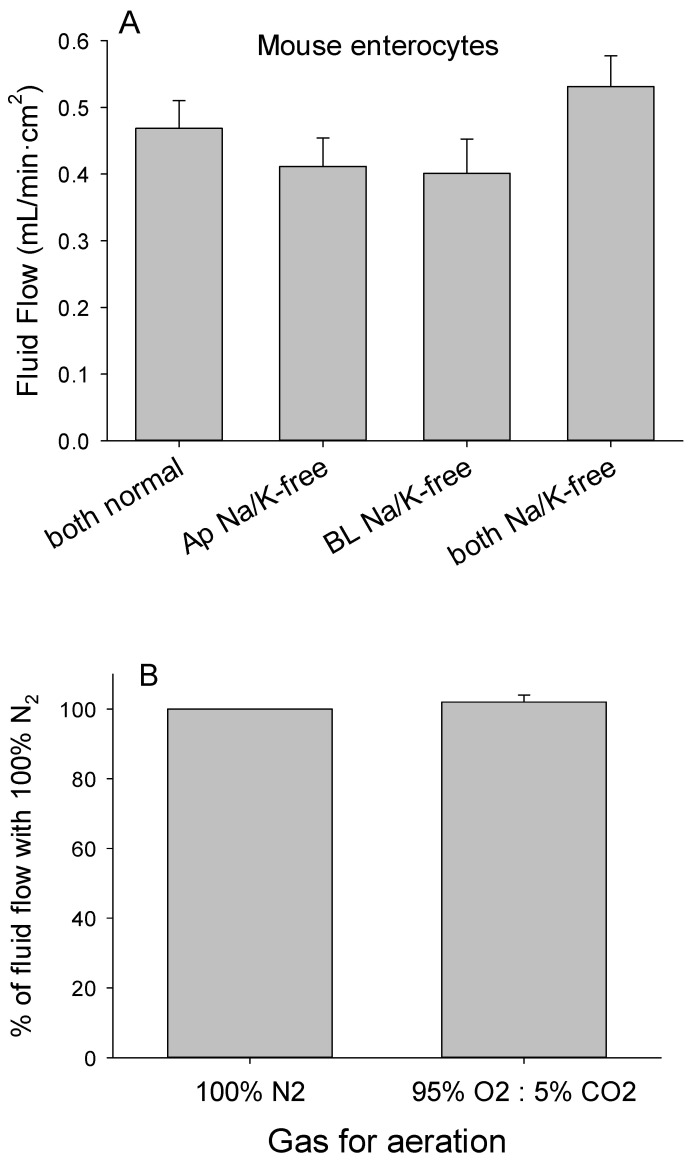
(**A**) Fluid flow by epithelia of mouse enterocytes cultured on inserts with 12 µm pores and exposed to vigorous aeration did not differ when both chambers were filled with normal Ringer’s solution (both normal) and when Na-free and K-free Ringer’s solution was added to the apical (Ap Na/K-free) or basolateral (BL Na/K-free) chamber or to both chambers (both Na/K-free). (**B**) Fluid flow by epithelia of enterocytes cultured on inserts with 12 µm pores did not differ when exposed to vigorous aeration using 100% nitrogen or the mixture of 95% oxygen and 5% CO_2_.

**Figure 9 medsci-09-00009-f009:**
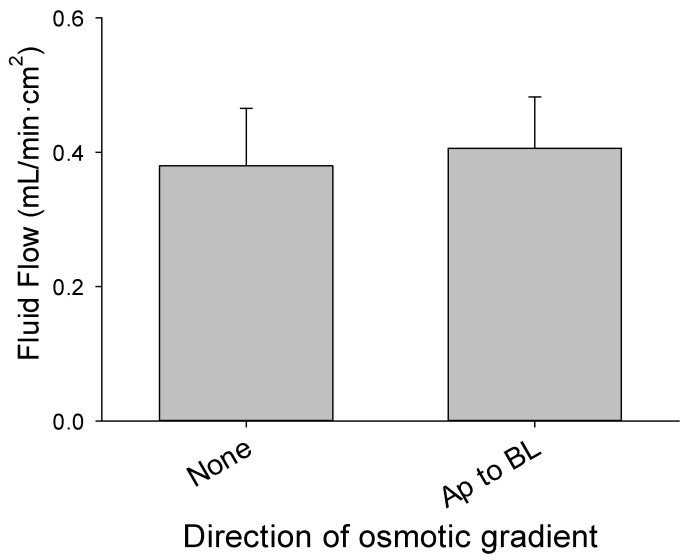
Basolateral-to-apical fluid flow by cultured enterocytes did not differ when measured with 290 mosmol normal Ringer’s solution in both chambers and when the Ringer’s solution in the apical chamber was diluted to 200 mosmol to create an apical-to-basolateral osmotic gradient (*n* = 4 inserts per condition).

**Figure 10 medsci-09-00009-f010:**
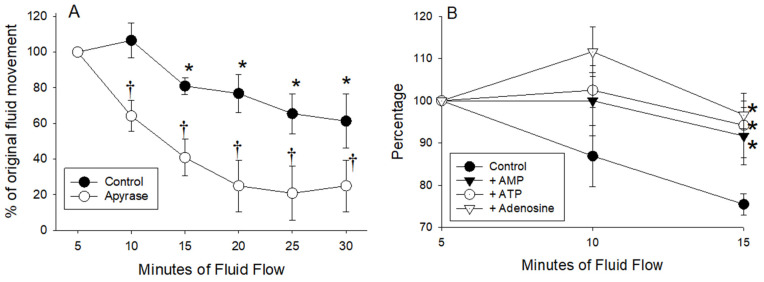
(**A**) Enterocytes harvested from two mice were cultured on inserts (*n* = 6 per mouse) and fluid flow in the presence and absence of apyrase was recorded for six consecutive 5 min periods with vigorous aeration. Fluid levels were equilibrated between each 5 min period. Values are expressed as a percentage of the fluid movement recorded for the first 5 min interval before the addition of apyrase. Differences are indicated for comparisons with initial readings (*) and for comparisons with and without apyrase (†). (B) The decline in rates of flow by control inserts were slowed by the addition of 0.5 mmol ATP, AMP and adenosine to both chambers. Values are expressed as the percentages of initial values. (* *p* < 0.05 for comparison with control tissues not exposed to ATP, AMP, or adenosine).

**Figure 11 medsci-09-00009-f011:**
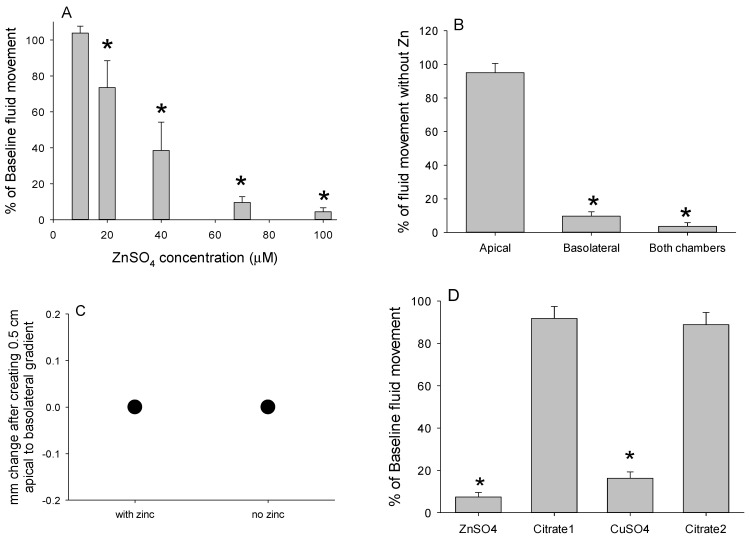
(**A**) Enterocytes harvested from 2 mice and cultured on inserts with 12 µm pores (*n* = 12 per mouse) were exposed to concentrations of zinc sulfate in both chambers ranging from 0 to 100 µM (*n* = 2 per concentration for each mouse). Values are expressed as the percentages of baseline flow prior to the addition of zinc (100%) and asterisks indicate a significant decline was detected (*p* < 0.05). (**B**) Enterocytes from 2 mice were cultured on inserts with 12 µm pores (*n* = 6 per mouse) and exposed to zinc sulfate (100 µM) only in the apical or the basolateral chamber, or in both chamber (*n* = 2 per condition for inserts prepared from each mouse). Values are the percentages of fluid flow prior to the addition of zinc and asterisks indicate a significant decline was detected (*p* < 0.05). (**C**) A basolateral-directed hydrostatic pressure gradient of 0.5 cm did not dissipate over a 15 min period in the presence or absence of zinc in both chambers (100 μM). (**D**) Enterocytes from two mice were cultured on inserts with 12 µm pores and exposed to vigorous aeration before adding either zinc or copper (100 µM) to inhibit fluid flow (*n* = 4 inserts per inhibitor per mouse) and then to citrate (50 μM), which restored fluid flow. Values are expressed as the percentage of the initial fluid flow that was significantly reduced by the addition of zinc sulfate or copper sulfate (*) and the subsequent recovery to near initial rates after the addition of sodium citrate.

**Figure 12 medsci-09-00009-f012:**
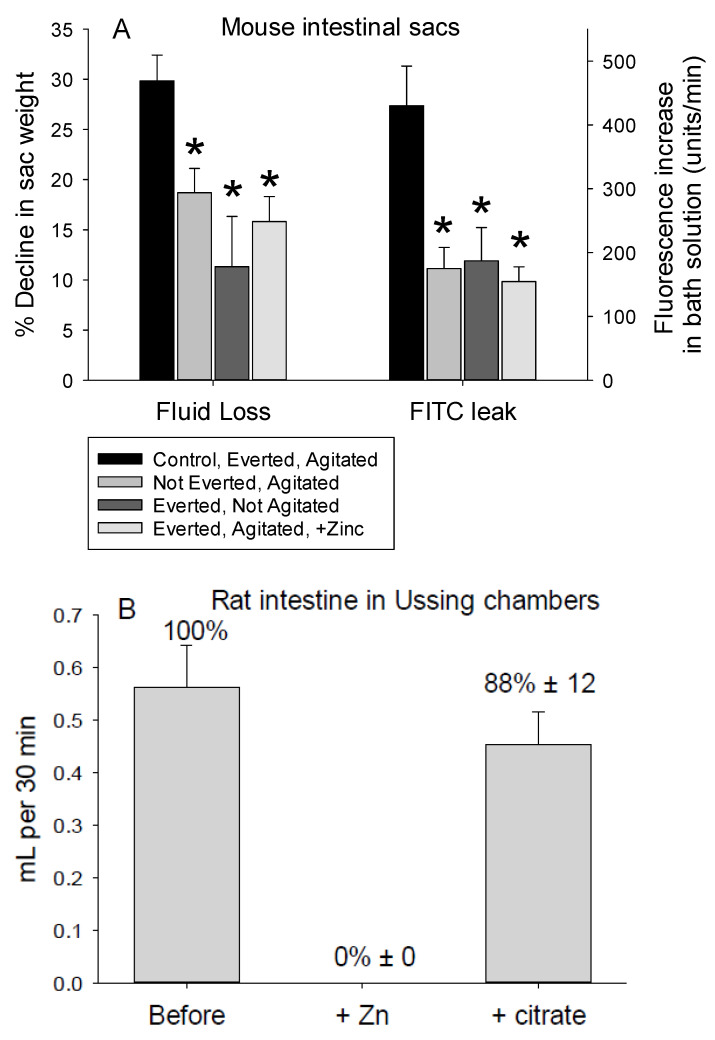
Unidirectional and inhibitable movement of fluid and 4 kD FITC-dextran is measurable with intact tissues. (**A**) Percentage decreases in weight, a measure of fluid loss (left group of bars) and leakage of 4 kD FITC-dextran into the Ringer’s solution incubation solution (right group of bars) from sacs of mouse small intestine as a function of whether the mucosa was internal or external, exposed to vigorous agitation by aeration and stirring, and the addition of zinc sulfate to the incubation solution. Sacs were prepared from 8 mice, with 4 sacs per mouse. (**B**) Fluid flow through intact small intestine from two rats mounted in Ussing chambers and exposed to vigorous aeration was measured before first the addition of zinc sulfate (100 μM) and then citrate (50 μM). Each period of measurements was 5 min (*n* = 4 preparations per rat). The asterisks indicate a significant difference was detected for comparisons with control everted sleeves.

**Figure 13 medsci-09-00009-f013:**
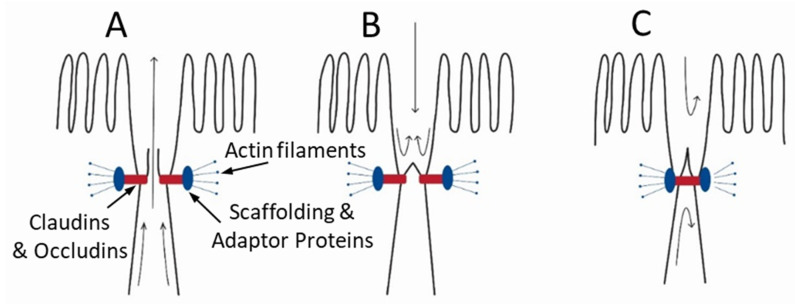
Hypothetical model of a regulated channel in the tight junction complex responsible for unidirectional filtration secretion of fluid into the lumen of the small intestine. (**A**) In the open state and with mechanical force, the tight junction pathway allows for unidirectional flow of fluid from the intercellular space into the lumen. (**B**) In a transient closed state, the reversal of flow from the lumen is prevented by an undefined mechanism that mimics a ‘one-way valve’. (**C**) In a stable closed (inactive) state, a conformational change in the tight junction pathway induced by zinc and other signals restricts fluid movement in either direction.

**Table 1 medsci-09-00009-t001:** Inserts with different pore sizes and without cells or epithelia were mounted in the Ussing system (*n* = 3 inserts per pore size) and both chambers were filled to the same level with Ringer’s solution. After an additional 0.5 mL of Ringer’s solution was added to the basolateral chamber, both chambers were aerated vigorously. The time required for the two chambers to equilibrate was recorded as an indicator of fluid flow and revealed that larger pores allow for greater flow.

Pore Size (µm)	Time to Equilibrate (min)
0.4	15.35 ± 0.61
3	5.12 ± 0.09
8	1.88 ± 0.07
12	1.00 ± 0.04

**Table 2 medsci-09-00009-t002:** Compounds used to alter transepithelia fluxes of chloride, sodium, and potassium by harvested enterocytes cultured on inserts with 12 μm pores. The known actions and final concentration in both chambers are provided. The change in fluid flow with vigorous aeration was based on comparisons of flows rates before and after addition of the inhibitors.

Compounds	Action	Conc.	Change in Flow
Glibenclamide	Inhibitor of CFTR and ATP-sensitive potassium channels (KATP)	200 µM	None
CFTRinh-172	Specific inhibitor of CFTR	20 µM	None
Bumetanide	Inhibitor of the Na/K/2Cl cotransporter in the basolateral membrane (NKCC1)	50 µM	None
Furosemide	Inhibitor of the Na/K/2Cl cotransporter	200 µM	None
Nystatin	Permeabilizes both the apical and basolateral membranes to cations	500 IU/mL	None
Ouabain	Inhibitor of Na-K-ATPase	1 mM	None
Amilioride	Inhibitor of the epithelial sodium channel, ENaC	50 µM	None
4,4′-Diisothiocyano-2,2′-stilbenedisulfonic acid (DIDS)	Inhibitor of the chloride–bicarbonate exchange AE1	100 µM	None
Tetraethylammonium	Potassium channel blocker	5 mM	None
Orthovandadate	Generalized inhibitor of phosphatase	100 µM	None

**Table 3 medsci-09-00009-t003:** Experimental evidence from inserts and cultured small intestinal epithelia that demonstrates unidirectional apically directed fluid flow is not an artifact of aeration.

Approach	Evidence
Expose naked inserts to aeration	There is no flow or development of a hydrostatic pressure gradient
Reverse orientation of epithelia	Direction of flow remains basolateral to apical
Establish hydrostatic pressure gradients	Apical-to-basolateral gradient persists, basolateral-to-apical gradient gradually dissipates
Fluid flow with Caco-2 cells	Lower flow, reversal of flow with chloride gradient
Expose fibroblasts to paradigm	No flow or development of hydrostatic pressure gradient; tight junctions are necessary
Addition of zinc then citrate	Inhibition of flow by zinc is reversed by citrate

## Data Availability

All of the data supporting the findings and conclusions of this study are presented in the results.

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
