# Peer review of "Paracellular Filtration Secretion Driven by Mechanical Force Contributes to Small Intestinal Fluid Dynamics"

_medsci, 2021, doi:10.3390/medsci9010009_

Round 1
Reviewer 1 Report
I believe the manuscript now has been corrected and fulfils this reviewer's comments and suggestions and thus can be accepted for publication.
One minor misstake remains on line 222 (table 1 text); remove Ringers two times!
Author Response
Thank you for bringing this mistake to our attention. It has been corrected. Also, thank you for your other comments and suggestions throughout this process. This has helped us to improve our contribution.
Reviewer 2 Report
The authors have done a detailed exploration of an incidental observation in the best tradition of scientific advances. I do not question the observations, but do have concerns about the extrapolation to intact animal physiology and pathophysiology. The authors suggest that filtration is an important pathophysiological event; I'm not sure that they made this case.
1) It is a big jump from a reconstituted cell layer on an artificial membrane to a normal mucosa in vivo which has nerves, lymphatics, and a tendency to increase pressure/tension on both sides of the mucosa with contraction of the muscularis causing pressurization because the mucosa is deformable and not fixed like a cell layer on a substrate in an Ussing chamber.
2) There is no measure of what the "mechanical force" is. The agitation due to bubbling is described in categories (none, low, vigorous). How much force is exerted on the cell layer? Alternatively, what pressure gradient existed between the two sides of the chamber? What happens if both sides are aerated vigorously?
3) How does the "mechanical force" in the experiment compare to the "mechanical force" in pathophysiological states in intact animals or people? My guess is that you do not know and so extrapolation of the experimental situation to pathophysiology is very speculative.
4) I like Figure 13 and suggest that the sidedness to the effect (B-L-->A, but not A-->B-L) is due to the geometry of the basolateral spaces. They resemble the Bay of Fundy in Canada where the tides get quite extreme as fluid is forced into an ever-narrowing estuary.
My advice is to report your experimental findings and cut out all the extrapolation to disease states until there is more information to support your speculations.
Author Response
The authors have done a detailed exploration of an incidental observation in the best tradition of scientific advances. I do not question the observations, but do have concerns about the extrapolation to intact animal physiology and pathophysiology. The authors suggest that filtration is an important pathophysiological event; I'm not sure that they made this case.
We appreciate the comments of the reviewer and we have given their concerns serious consideration in revising our contribution. We agree with the reviewer’s concern about the limitations of extrapolating in vitro data to intact animal physiology and pathophysiology. We have revised language in the discussion accordingly. However, it is reasonable to conclude from our data that filtration secretion contributes to the relationship between intestinal motility and fluid secretion, which cannot be explained solely by chloride secretion.
1) It is a big jump from a reconstituted cell layer on an artificial membrane to a normal mucosa in vivo which has nerves, lymphatics, and a tendency to increase pressure/tension on both sides of the mucosa with contraction of the muscularis causing pressurization because the mucosa is deformable and not fixed like a cell layer on a substrate in an Ussing chamber.
We agree, and therefore confirmed the Ussing chamber findings using intact mouse and rat intestines (Study 7 and Study 8) to determine if what we observed with the reconstituted cell layer on an artificial membrane could be replicated by tissues with an intact mucosa.
We recognize a limitation of in vitro studies is how difficult it is to replicate in situ physiology and conditions, which include contraction of the muscle layers. Still, in vitro studies are essential for understanding physiological processes. An example of this can be drawn from the decades of in vitro studies of intestinal nutrient absorption and ion secretion that have used Ussing chambers and intact intestine, as we have done in this study. The findings from the in vitro absorption and secretion studies have proven invaluable for understanding in vivo mucosal functions in health and disease. We contend our in vitro findings similarly provide novel insights into the mucosal function of fluid secretion that does not rely on chloride secretion.
To clarify this even more, we include in the Discussion how our observations of unidirectional movement of fluid correspond with the early descriptions of unidirectional filtration secretion by intact small intestine in response to induced changes in pressure and to the contraction state of the smooth muscle.
2) There is no measure of what the "mechanical force" is. The agitation due to bubbling is described in categories (none, low, vigorous). How much force is exerted on the cell layer? Alternatively, what pressure gradient existed between the two sides of the chamber? What happens if both sides are aerated vigorously?
We did not quantify the mechanical force imposed on the cultured cells or the intact small intestine tissues exposed to stirring and aeration. We are not aware of how this can be done and therefore used a qualitative measure. This does not diminish the fact the amount of mechanical force imposed on the intestinal epithelium influences the rate of filtration secretion.
As described on line 117 of the reviewed manuscript, both chambers were filled to the same level to avoid an initial hydrostatic pressure gradient to examine the influence of different hydrostatic pressure gradients.
As described on line 229, both chambers were aerated at the same bubbling intensity except when we specifically aerated one or the either side to determine if this made a difference.
3) How does the "mechanical force" in the experiment compare to the "mechanical force" in pathophysiological states in intact animals or people? My guess is that you do not know and so extrapolation of the experimental situation to pathophysiology is very speculative.
To address the reviewer’s concern we have included in the final section a need to quantify both the experimental and the in situ mechanical forces imposed on the mucosa to better understand the magnitude of filtration secretion in vivo and we have softened the language throughout.
Although gut motility is known to impose mechanical force on the mucosa and the associated epithelium, to our knowledge there are no measures of the actual mechanical force intestinal motility imposes on the mucosa and epithelium. Therefore, even if we were able to quantify the mechanical force we imposed on the epithelia and intact tissues, we have no way of knowing how this compares to in situ mechanical force. Perhaps the reviewer is aware of published (or unpublished) data?
4) I like Figure 13 and suggest that the sidedness to the effect (B-L-->A, but not A-->B-L) is due to the geometry of the basolateral spaces. They resemble the Bay of Fundy in Canada where the tides get quite extreme as fluid is forced into an ever-narrowing estuary.
The reviewer makes a nice observation and provides a clever analogy. Unlike the Bay of Fundy, filtration secretion is unidirectional and the lateral space between two adjacent enterocytes (the Bay) can collapse and re-open (empty and refill).
My advice is to report your experimental findings and cut out all the extrapolation to disease states until there is more information to support your speculations.
We have revised the discussion to put our findings into context with what is known about the relationship between gut motility and the volumes of fluid that enter the intestine in disease states and have softened the language to avoid conclusive statements. We also indicate in the concluding section of the need to obtain more information to understand how our findings pertain to normal and disease states.
Round 2
Reviewer 2 Report
The last two sentences of the abstract should be revised to reflect the softening of the proposed pathophysiological mechanism in the Conclusion because of a lack of supporting evidence.
Existing: The role of intestinal motility as a driving force for filtation secretion in vivo is consistent with the pathophysiology of diarrhea and explains why the most effective antidiarrheal agents reduce smooth muscle contraction. This pathway provides a promising avenue for management of diarrhea and constipation.
Proposed: The role of filtration secretion in the genesis of diarrhea in intact animals needs further study; our findings may explain a potential linkage between intestinal motility and intestinal fluid transport.
Author Response
We thank the reviewer for the suggested change which we accepted.
This manuscript is a resubmission of an earlier submission. The following is a list of the peer review reports and author responses from that submission.
Round 1
Reviewer 1 Report
The manuscript titled "Paracellular Filtration Secretion Driven by Mechanical Force Contributes to Small Intestinal Fluid Dynamics" deals with the controversial topic of the paracellular transport of water into the lumen. The authors describe a process whereby the agitation of epithelia by vigorous aeration induces water transport from the basolateral to the apical side of an Ussing chamber to develop a head difference between the two sides. They postulate that the pressure fluctuations due to the aeration push the water through one-way valves associated with the tight junctions in a manner analogous to a reciprocating pump, e.g. a diaphragm pump.
The findings are interesting as they point to a mechanism that in combination with other accepted mechanisms could explain the observed flux of bidirectional flow of water across the epithelia. The series of experiments described has been designed well enough to convince me that this is a real phenomenon and not merely some form of experimental artefact. Perhaps one weakness is that I know of no morphological evidence to back up Figure 14.
Other Points:
Line 112: Replace “City and State” with actual location.
Line 118: Correct chemical formula “KH8PO4”.
Line 206: I don’t understand why the relationship between flow and pore size was “unexpected”. For interest, the equation on Line 209 is for the flow through a single pore and if you assume that the number of pores per unit area scales roughly with 1/R2 then the flux (volumetric flow/unit are) is proportional to R2.
Lines 294-295: While the experiment gives a good indication that tight junctions are necessary, I don’t think it necessarily meets the level of “proof” in that the experiment has changed two variables at the same time, i.e. cell type and the extent of tight junction development.
Lines 613-616: In order to say that the findings of this work “indicate” that “filtration secretion driven by the increased motility plays a major role” one must first show that the effect of vigorous aeration is physiologically comparable to that of gut motility, which I don’t think this work has attempted. This is a hypothesis that requires further investigation.
Author Response
Comments and Suggestions for Authors
The manuscript titled "Paracellular Filtration Secretion Driven by Mechanical Force Contributes to Small Intestinal Fluid Dynamics" deals with the controversial topic of the paracellular transport of water into the lumen. The authors describe a process whereby the agitation of epithelia by vigorous aeration induces water transport from the basolateral to the apical side of an Ussing chamber to develop a head difference between the two sides. They postulate that the pressure fluctuations due to the aeration push the water through one-way valves associated with the tight junctions in a manner analogous to a reciprocating pump, e.g. a diaphragm pump.
The findings are interesting as they point to a mechanism that in combination with other accepted mechanisms could explain the observed flux of bidirectional flow of water across the epithelia. The series of experiments described has been designed well enough to convince me that this is a real phenomenon and not merely some form of experimental artefact. Perhaps one weakness is that I know of no morphological evidence to back up Figure 14.
We thank the reviewer for their comments and accurate synopsis of the study. Pertaining to the weakness mentioned, although paracellular movement of fluid and solutes and the roles of various tight junction proteins have been described numerous times, morphological evidence remains elusive. Our functional data for a regulated water channel in the tight junction complex should foster studies that will eventually provide morphological and molecular evidence for Figure 14.
Other Points:
Line 112: Replace “City and State” with actual location. Done- thank you for seeing this oversight.
Line 118: Correct chemical formula “KH8PO4”. We thank the reviewer for seeing this typo. The correct chemical formula of KH2PO4 is now used.
Line 206: I don’t understand why the relationship between flow and pore size was “unexpected”. For interest, the equation on Line 209 is for the flow through a single pore and if you assume that the number of pores per unit area scales roughly with 1/R2 then the flux (volumetric flow/unit are) is proportional to R2. The sentence has been shortened by removing the statement about being unexpected. The reviewer’s comment about the equation being for a single pore is correct. As noted, the flow through the entire insert would reflect both pore size and the number of pores.
Lines 294-295: While the experiment gives a good indication that tight junctions are necessary, I don’t think it necessarily meets the level of “proof” in that the experiment has changed two variables at the same time, i.e. cell type and the extent of tight junction development. We agree definitive proof is lacking and have changed the wording from “tight junctions are” to “tight junctions appear to be”.
Lines 613-616: In order to say that the findings of this work “indicate” that “filtration secretion driven by the increased motility plays a major role” one must first show that the effect of vigorous aeration is physiologically comparable to that of gut motility, which I don’t think this work has attempted. This is a hypothesis that requires further investigation. We agree that epithelia mounted in Ussing chambers and exposed to vigorous aeration is different from the mechanical force associated with gut motility acting in vivo on the epithelium of the intact small intestine. We used aeration and stirring as a substitute for motility that would impose force on the mucosa. We exposed rat and mouse intestine to conditions that moved the mucosa to learn if this would replicate our findings with cultured enterocytes. We also agree that our findings require further investigation to understand more about how gut motility in vivo drives fluid through the tight junction into the gut lumen. What is not to be overlooked is the reversible inhibitory influence of zinc in each of the experimental paradigms.
Reviewer 2 Report
Reviewer’s comments and suggestions on the paper Paracellular filtration…., by R.K. Buddington et al (medsci-939238).
Major comments:
This is a very interesting paper highlighting intestinal fluid secretion by presenting in vitro results on filtration secretion driven by mechanical force. The authors results indicate that mechanical force generated by vigorous aeration/agitation of intestinal epithelial monolayers as well as intestinal mucosal preparations, can drive fluid secretion through the tight junctions in the baso-lateral to apical direction. The results of this study are of importance for explaining the large volumes of fluid secretion in the small intestine generated during digestion and especially diarrhoea and linked to increased intestinal motility. Based on their findings the authors present a model of regulated channels in the tight junctions mediating the filtration secretion into the small intestinal lumen.
The way of producing the mechanical force and thus a hydrostatic pressure gradient by vigorous aeration seems a bit ambiguous, i.e., tough treatment of the specimens, and the results would be even more convincing if the hydrostatic pressure gradient could be generated also in an alternative way. Actually, an attempt was done, as presented in Fig. 6, by generating a hydrostatic pressure gradient over the enterocyte monolayer by a increase the liquid height on the baso-lateral side to the apical side. If one compares the 5 min fluid displacement of the 10 mm difference of the baso-lateral to apical side, as shown in the figure, it corresponds to about 30 % of that generated by aeration. Consequently, if one increases the basal to apical liquid height about 3-4 times (i.e., to 30-40 mm) would it mean that the fluid displacement would increase to about the same as during 5 min of aeration and thus reflect the hydrostatic pressure generated by vigorous aeration in your system? Moreover, what would happen if you increase the hydrostatic pressure by increasing the liquid height on both sides of the Ussing chambers? Is it of importance that the hydrostatic pressure possibly fluctuates due to the aeration/bubbling?
Minor comments:
Line 152 etc (incl. Figure 13 text); It is misleading to call the intestinal tissue preparations intact tissue since they were stripped with blood and nerve supply as well as serosa removed!
Lines 118, 141, 188 etc; The authors should avoid the laboratory slang Ringers and instead use Ringer’s solution.
Line 220; The results presented in Fig. 2 could preferentially be presented in a more space-saving table instead.
Line 16, 55, 531 etc and Table 2; Shouldn’t it be hydrostatic pressure gradient?
Study 7 and 8; In this reviewer’s experience intestinal specimens are fragile with e.g. villi loosening during longer in vitro incubations and tough handling (aeration). As a measure of tissue viability after treatment it would have been interesting to see histology preparations of the intestines after incubation with vigorous agitation up to 90 min.
Line 706; inserts with 12 µm.. (in line with earlier text).
Author Response
Comments and Suggestions for Authors
Reviewer’s comments and suggestions on the paper Paracellular filtration…., by R.K. Buddington et al (medsci-939238).
Major comments:
This is a very interesting paper highlighting intestinal fluid secretion by presenting in vitro results on filtration secretion driven by mechanical force. The authors results indicate that mechanical force generated by vigorous aeration/agitation of intestinal epithelial monolayers as well as intestinal mucosal preparations, can drive fluid secretion through the tight junctions in the baso-lateral to apical direction. The results of this study are of importance for explaining the large volumes of fluid secretion in the small intestine generated during digestion and especially diarrhoea and linked to increased intestinal motility. Based on their findings the authors present a model of regulated channels in the tight junctions mediating the filtration secretion into the small intestinal lumen.
We appreciate the comments of the reviewer and their recognition of the importance of our findings for gaining a better understanding of intestinal fluid dynamics.
The way of producing the mechanical force and thus a hydrostatic pressure gradient by vigorous aeration seems a bit ambiguous, i.e., tough treatment of the specimens, and the results would be even more convincing if the hydrostatic pressure gradient could be generated also in an alternative way. Actually, an attempt was done, as presented in Fig. 6, by generating a hydrostatic pressure gradient over the enterocyte monolayer by a increase the liquid height on the baso-lateral side to the apical side. If one compares the 5 min fluid displacement of the 10 mm difference of the baso-lateral to apical side, as shown in the figure, it corresponds to about 30 % of that generated by aeration. Consequently, if one increases the basal to apical liquid height about 3-4 times (i.e., to 30-40 mm) would it mean that the fluid displacement would increase to about the same as during 5 min of aeration and thus reflect the hydrostatic pressure generated by vigorous aeration in your system? Moreover, what would happen if you increase the hydrostatic pressure by increasing the liquid height on both sides of the Ussing chambers? Is it of importance that the hydrostatic pressure possibly fluctuates due to the aeration/bubbling? The reviewer raises several points that we address individually. They mimic some of our initial thoughts when we first observed the basal to apical movement of fluid across epithelia.
- As reviewer #1 commented, it is unknown how our approach of using vigorous aeration to impose force on epithelia prepared from cultured enterocytes compares with the forces imposed in vivo by gut motility.
- The reviewer is astute by noting how the basal to apical movement of fluid down a hydrostatic pressure gradient is increased by aeration, which is very true and is consistent with a one-way valve. As for the magnitude of the hydrostatic pressure gradient, we were limited by the size (height) of the chambers. It would be expected that the height of the basal to apical gradient is related to the rate of fluid movement from basal to apical. If both sides at the start are balanced (no pressure gradient), the initial height on both sides should not have an influence on the movement in response to aeration, as long as the height is the same. However, we did not directly test this.
- It is possible the aeration causes some rapid and transient fluctuations in hydrostatic pressure. However, these would likely be negligible compared with the gradient generated by the movement of fluid. If aeration alone causes a significant hydrostatic pressure gradient, this would have been measured by changes in volumes when we exposed naked inserts without cells to aeration.
Minor comments:
Line 152 etc (incl. Figure 13 text); It is misleading to call the intestinal tissue preparations intact tissue since they were stripped with blood and nerve supply as well as serosa removed! For our studies with rat intestine we removed only the serosa, which would have included nerve and vasculature but not the underlying muscle layers, which is commonly done by other investigators before mounting the isolated mucosal tissue in the Ussing chambers. The mouse preparations were fully intact with the serosa. We revised the description to clarify what was done.
Lines 118, 141, 188 etc; The authors should avoid the laboratory slang Ringers and instead use Ringer’s solution. We have made the change as suggested.
Line 220; The results presented in Fig. 2 could preferentially be presented in a more space-saving table instead. We consider a figure rather than a table presents the data in a form that makes the differences between different pore sizes very apparent. However, we suggest the editor make the final decision.
Line 16, 55, 531 etc and Table 2; Shouldn’t it be hydrostatic pressure gradient? We thank the reviewer for this suggestion. We have corrected the manuscript accordingly.
Study 7 and 8; In this reviewer’s experience intestinal specimens are fragile with e.g. villi loosening during longer in vitro incubations and tough handling (aeration). As a measure of tissue viability after treatment it would have been interesting to see histology preparations of the intestines after incubation with vigorous agitation up to 90 min. The reviewer’s comment is appreciated. Although we did not perform histology of the tissues before and after the exposure to aeration, the ability of the tissue preparations to move fluid from basal to apical in the presence of agitation indicates if there was damage to the epithelium, it was not sufficient to stop the unidirectional movement of fluid. A number of years ago (in the 80’s) during studies of intestinal nutrient transport we were also concerned that exposing everted sleeves to aeration and vigorous stirring of the uptake solutions (1,200 rpm) would damage the epithelium. Histology revealed little if any damage. Moreover, the inhibition by zinc provides additional evidence the epithelium was still functional.
Line 706; inserts with 12 µm.. (in line with earlier text). We are not able to locate this issue in our manuscript.
Round 2
Reviewer 1 Report
Thank you for an interesting read.
Reviewer 2 Report
Comments and suggestions (2nd):
The few changes done in the manuscript are ok but still I believe that some of the comments made by this reviewer and answered in the authors could be incorporated in the manuscript to better show not only the benefits but also the drawbacks/shortcomings of the study. Possibly this could be done in a finishing paragraph in the manuscript.
Another question that came to this reviewer’s mind is what will happen if also the outer muscularis layer (or both) is removed during the stripping procedure – according to the authors theory this would interfere with the generation of the hydrostatic pressure gradient over the intestinal mucosa.
Maybe I wasn’t clear enough in my comment and motivation but the results presented in fig. 2 are more of a technical art than of biological significance and thus could preferentially be presented in a space-saving table!